

# Assessing conservation status of resident and migrant birds on Hispaniola with mist-netting

John D. Lloyd, Christopher C. Rimmer and Kent P. McFarland

Vermont Center for Ecostudies, Norwich, VT, United States

## ABSTRACT

We analyzed temporal trends in mist-net capture rates of resident ($n = 8$) and overwintering Nearctic-Neotropical migrant ($n = 3$) bird species at two sites in montane broadleaf forest of the Sierra de Bahoruco, Dominican Republic, with the goal of providing quantitative information on population trends that could inform conservation assessments. We conducted sampling at least once annually during the winter months of January–March from 1997 to 2010. We found evidence of declines in capture rates for three resident species, including one species endemic to Hispaniola. Capture rate of Rufous-throated Solitaire (*Myadestes genibarbis*) declined by 3.9% per year (95% CL = 0%, 7.3%), Green-tailed Ground-Tanager (*Microligea palustris*) by 6.8% (95% CL = 3.9%, 8.8%), and Greater Antillean Bullfinch (*Loxigilla violacea*) by 4.9% (95% CL = 0.9%, 9.2%). Two rare and threatened endemics, Hispaniolan Highland-Tanager (*Xenoligea montana*) and Western Chat-Tanager (*Calyptophilus tertius*), showed statistically significant declines, but we have low confidence in these findings because trends were driven by exceptionally high capture rates in 1997 and varied between sites. Analyses that excluded data from 1997 revealed no trend in capture rate over the course of the study. We found no evidence of temporal trends in capture rates for any other residents or Nearctic-Neotropical migrants. We do not know the causes of the observed declines, nor can we conclude that these declines are not a purely local phenomenon. However, our findings, along with other recent reports of declines in these same species, suggest that a closer examination of their conservation status is warranted. Given the difficulty in obtaining spatially extensive, long-term estimates of population change for Hispaniolan birds, we suggest focusing on other metrics of vulnerability that are more easily quantified yet remain poorly described, such as extent of occurrence.

Corresponding author
John D. Lloyd,
jlloyd@vtecostudies.org,
5355693@gmail.com

## INTRODUCTION

Hispaniola supports a notably diverse avifauna, including at least 31 endemic species (*Latta et al., 2006*), several of which appear to be the only extant members of ancient, family-level clades (*Barker et al., 2013*; *Barker et al., 2015*). Many of these taxa are of substantial conservation concern given extensive habitat loss caused by ongoing deforestation

in both Haiti and the Dominican Republic (*Stattersfield et al., 1998*; *Latta, 2005*). None of the endemic birds of Hispaniola have been well studied, however, and assessments of their conservation status are often qualitative, subjective, and based largely on expert opinion (*Latta & Fernandez, 2002*). Decisions about investments in conservation are often guided by population status (*Possingham et al., 2002*; *Rodrigues et al., 2006*), and thus well-informed status assessments are critically important for the effective allocation of limited funding for conservation.

Here, we seek to improve current understanding of the conservation status of the unique and threatened assemblage of birds in montane cloud forest in Sierra de Bahoruco, Dominican Republic. These forests are a hotspot of endemism on the island (*Latta, 2005*), support several globally threatened resident bird species, and constitute a principal wintering area for the globally Vulnerable (*BirdLife International, 2012*) Bicknell's Thrush (*Catharus bicknelli*), a Nearctic-Neotropical migrant. Montane cloud forests also face substantial and ongoing threats from deforestation for agricultural production and expansion of human settlements, even in ostensibly protected areas such as Sierra de Bahoruco National Park (*BirdLife International, 2015*). We used data collecting during 13 years of mist-netting at two different sites to estimate temporal trends in capture rate, which we use as an index of change in population size and as a means to draw inference about conservation status. In other tropical systems, long-term mist netting has proven a useful tool for identifying population declines in bird assemblages that are otherwise difficult to monitor (e.g., *Faaborg et al., 2013*).

## METHODS

From 1995 to 2010, we operated a standardized array of 30–35 mist nets (we used 6- and 12-m nets that were 2.6-m tall with 36-mm mesh) at two remote sites in Sierra de Bahoruco, southwestern Dominican Republic. The sites, Pueblo Viejo (hereafter "PUVI"; 18.2090°N, −71.5080°W) and Palo de Agua (hereafter "PALO"; 18.2047°N, −71.5321°W), consist of montane cloud forest at 1,775–1,850 m elevation in Sierra de Bahoruco National Park and are separated by 2.6 km of contiguous forest. Both sites are characterized by a dense understory composed largely of thick woody tangles, complete broadleaf canopy cover with trees reaching heights between 15 and 20 m, and an abundance of lianas and epiphytes (*Veloz, 2007*). We trimmed vegetation to prevent overgrowth of our net lanes, but otherwise the forest at both sites was undisturbed by humans.

We visited PUVI at least once annually between November and May, except in 1999 when we did not visit either site. We made two visits in 1997 (March, November), 1998 (March, November), 2002 (February, May), 2003 (February, May), and 2010 (March, November). Our one visit to PUVI in 1996 occurred in early December. To minimize the potentially confounding effects of seasonal variation in abundance and bird behavior that may affect capture rate, this analysis does not include data collected during the May, November, and December visits. Resident birds have commenced breeding by May, and so availability for capture may be different during this period. Transient hatch-year birds,

which likely have a very different probability of capture, also begin appearing in large numbers in May. Migrant birds have departed for their breeding grounds by May, but are still arriving at our sites during November and, to a lesser extent, early December.

We also visited PALO at least once annually during the same period that we visited PUVI, except for 1996 and 1999–2001. As with PUVI, we excluded data collected during the three November visits (1997, 1998, and 2010). We did not visit PALO in May. At both sites in 1995, we only banded Nearctic-Neotropical migrants, and so we excluded data from that year from this analysis. The final, censored data set for this analysis thus includes captures made from 1997 to 2010 (with no data collected in 1999) on dates ranging from 24 January to 21 March. We believe that this date range reflects a period of relative stability at our sites, after migrant species have arrived, settled, and established winter territories but before resident species have commenced breeding. As such, we also believe that capture rates during this period are comparable among years because availability for and probability of capture should be relatively constant among years.

At each site, we established permanent net locations along three parallel foot trails 100–150 m apart. The area bounded by the foot trails was ~25 ha at PUVI and ~15 ha at PALO. We regularly used 30 net locations at PUVI and 35 at PALO. Nets were typically operated for 3 days at each site, beginning in late afternoon of day 1, from dawn to dusk on days 2 and 3, and until mid-morning on day 4. Nets were checked hourly and closed under adverse weather conditions. We recorded daily opening and closing times of each net. Both sites were netted in succession each year, with set-up at the second site occurring on the day that nets were removed from the first site.

We placed US Fish and Wildlife Service aluminum leg bands on all Nearctic-Neotropical migrant species and custom-made, uniquely numbered leg bands (Gey Band and Tag Company) on all Hispaniolan resident species, except for Hispaniolan Emeralds (*Chlorostilbon swainsonii*), which was too small for our bands. We aged and sexed all North American species using standard criteria according to *Pyle (1997)* and all resident species using criteria available in field guides (*Latta et al., 2006*) or based on our own accumulated field knowledge. However, we could only reliably age and sex a handful of species, so we pooled capture rates for all ages and all sexes in our analyses.

We analyzed trends in capture rate for 6 endemic species that we believed were adequately sampled by our methods (English common names follow *Latta et al. (2006)*, scientific names follow *AOU (2015)*: Narrow-billed Tody (*Todus angustirostris*), Green-tailed Ground-Tanager (*Microligea palustris*), Hispaniolan Highland-Tanager (*Xenoligea montana*), Black-crowned Palm-Tanager (*Phaenicophilus palmarum*), Western Chat-Tanager (*Calyptophilus tertius*), and Hispaniolan Spindalis (*Spindalis dominicensis*)). Quantitative data on population trends are lacking for all of these species, but two are suspected of being at risk of extinction: Hispaniolan Highland-Tanager is recognized as Vulnerable on the IUCN Red List (*BirdLife International, 2012*) and Endangered by *Latta et al. (2006)*, and Western Chat-Tanager is considered Critically Endangered by *Latta et al. (2006)*. BirdLife International does not recognize the taxonomic separation of Eastern Chat-Tanager (*C. frugivorous*) and Western Chat-Tanager, and instead considers the entire species group Vulnerable (*BirdLife International, 2012*).

We also analyzed standardized capture rates for the two most common non-endemic residents (Rufous-throated Solitaire (*Myadestes genibarbis*) and Greater Antillean Bullfinch (*Loxigilla violacea*)) and the three most frequently encountered North American migrants at our sites: Bicknell's Thrush, Ovenbird (*Seiurus aurocapilla*), and Black-throated Blue Warbler (*Setophaga caerulescens*). All of the resident species that we captured breed regularly at both sites (*Rimmer et al., 2008*; CC Rimmer, 2015, unpublished data).

We assumed that the number of captures of each species could be approximated by the Poisson distribution and used a generalized linear model to examine temporal and spatial trends in capture rate among species. The response variable was the number of unique individuals (new bandings and returns from previous sessions, but not repeat captures from the same session) of each species captured during each unique capture session (hence two data points for PUVI in 1998, when we visited in both February and March). We accounted for variation in capture effort by using the number of net hours per capture session (log-transformed) as an offset in the model. We calculated net hours by multiplying the number of 12-m mist nets (or their equivalent; e.g., a 6-m net open was equivalent to a 0.5 12-m net) in use during each session by the length of time each was open. For the purposes of standardization with other constant-effort mist-netting studies, we report capture rate per 1,000 net hours (i.e., expected captures for every 1,000 h that 12-m net was open). The predictor variables included year, site, and the interaction between site and year. We considered three models for each species: year only, site + year, and site∗year. We chose among these competing models with a likelihood-ratio test. We estimated temporal trends using the estimated coefficient for the year effect in the best model, and established an a priori significance level of $\alpha = 0.05$.

Once we identified the best model, we examined whether we could further improve model fit by adding to the best model a parameter reflecting the average multivariate El Niño-Southern Oscillation (ENSO) index (MEI) during June to December prior to each banding session. We used lagged values from the previous June to December because they provided a measure of the relative strength of the ENSO event and thus the potential influence on rainfall during the wet and dry seasons preceding our banding sessions. As they build in strength, warm ENSO events are associated with anomalously dry conditions during the late wet season (September–October) and most of the subsequent dry season (November–March), and with anomalously wet conditions during the early wet season (April–July) of the following year as the event subsides (*Chen & Taylor, 2002*). We downloaded bimonthly MEI values from http://www.esrl.noaa.gov/psd/enso/mei/table.html for the June to December prior to each banding session, and averaged these values to produce a single average value for those six months, which we then added as a covariate to the best-fitting model. We determined whether addition of the MEI covariate improved model fit via a likelihood-ratio test.

Examination of residual plots and QQ-plots did not reveal any deviations from model assumptions regarding the distribution of residuals or the relationship between residuals and fitted values. For each model, we calculated the autocorrelation of residuals at each possible time lag, and compared it to expectations under the null hypothesis of no

autocorrelation. We found no evidence of autocorrelation in residuals. We also tested the null hypothesis of no autocorrelation among residuals by regressing the value of each residual against its lagged $(t-1)$ value; in no case could we reject the null hypothesis (all $P$ values $> 0.05$). We used the ratio of the residual deviance to the deviance degrees of freedom as a measure of overdispersion. We found little evidence of overdispersion (residual deviance $<2$ times the residual degrees of freedom), so we made no adjustment to the models (although we note that quasi-Poisson and negative binomial models produce results that do not differ qualitatively from the Poisson). We report pseudo-$R^2$ as an approximate measure of the explanatory power of the best model in each analysis, calculated as: $1 - \frac{\text{residual deviance}}{\text{null deviance}}$.

All analyses were conducted using R (*R Core Team, 2015*). All data used in this analysis are available in *Lloyd et al. (2015)*. Permission to band North American migrants was granted by the USGS Bird Banding Lab, under a permit issued to CCR (permit no. 23,541), and research activities in the Dominican Republic were approved by the Subsecretaria de Áreas Protegidas y Biodiversidad.

## RESULTS

We conducted 15 banding sessions at PUVI over 13 years and 11 banding sessions over the same period at PALO (Table 1), yielding $>22,000$ net hours. We captured a total of 31 species (Table 2). The endemic Green-tailed Ground-Tanager was the most commonly encountered species; number of captures for species included in this analysis ranged from 69 to 245 individuals (Table 2).

Capture rates declined over the course of our study for Rufous-throated Solitaire ($\beta_{\text{year}} = -0.04$, 95% CL $= -0.076, -0.001$; $P = 0.04$; Fig. 1), Green-tailed Ground-Tanager ($\beta_{\text{year}} = -0.07$; 95% CL $= -0.092, -0.040$; $P < 0.001$; Fig. 2), and Greater Antillean Bullfinch ($\beta_{\text{year}} = -0.05$; 95% CL $= -0.097, -0.009$, $P = 0.02$; Fig. 3). These estimated coefficients equate to expected annual declines in the number of captures of 3.9% (95% CL $=0\%$, 7.3%) for Rufous-throated Solitaire, 6.8% (95% CL $= 3.9\%$, 8.8%) for Green-tailed Ground-Tanager, and 4.9% (95% CL $= 0.9\%$, 9.2%) for Greater Antillean Bullfinch. Capture rate also varied by site for these species; for Rufous-throated Solitaire, expected counts were higher at PALO (Fig. 1), whereas for Green-tailed Ground-Tanager and Greater Antillean Bullfinch counts were greater at PUVI (Figs. 2 and 3). We found no evidence of a site-by-year interaction in capture rates for any of these species. The absolute magnitude of declines in expected capture rate over the course of the study were relatively small for Rufous-throated Solitaire and Greater Antillean Bullfinch. For Rufous-throated Solitaire, the decline in expected capture rate amounted to 4.5 fewer individuals captured per 1,000 net hours at PALO in 2010 as compared to 1997, and 2.7 fewer individuals per 1,000 net hours at PUVI over the same period of time. For Greater Antillean Bullfinch, the declines were even smaller: 1.6 fewer individuals per 1,000 net hours at PALO, and 4.8 fewer individuals per 1,000 net hours at PUVI. The magnitude of the decline in expected capture rate for Green-tailed Ground-Tanager was more substantial: expected capture rate in 2010 was 5.7 fewer individuals per 1,000 net hours than in 1997 at PALO, and 17.0 fewer individuals per 1,000 net hours at PUVI.

Table 1 **Summary of capture effort.** Dates of banding sessions and total net hours at two sites in the Sierra de Bahoruco, Dominican Republic from 1997 to 2010.

| Date | Site[a] | Net hours[b] |
|---|---|---|
| 1997 February 28–March 08 | PUVI | 1136.8 |
| | PALO | 743.1 |
| 1998 March 04–March 11 | PUVI | 947.0 |
| | PALO | 921.0 |
| 2000 January 24–January 27 | PUVI | 571.0 |
| | PALO | 0 |
| 2001 January 30–February 05 | PUVI | 793.0 |
| | PALO | 0 |
| 2002 February 10–February 17 | PUVI | 867.5 |
| | PALO | 941.0 |
| 2003 January 30–February 11 | PUVI | 919.5 |
| | PALO | 995.0 |
| 2004 February 19—February 28 | PUVI | 474.9 |
| | PALO | 971.5 |
| 2005 February 04–February 10 | PUVI | 940.5 |
| | PALO | 895.8 |
| 2006 January 26–January 31 | PUVI | 907.9 |
| | PALO | 674.8 |
| 2007 January 31–February 07 | PUVI | 1481.0 |
| | PALO | 1085.8 |
| 2007 March 13–March 17 | PUVI | 1239 |
| | PALO | 0 |
| 2008 February 07–February 12 | PUVI | 920.0 |
| | PALO | 591.5 |
| 2008 March 13–March 16 | PUVI | 711.5 |
| | PALO | 0 |
| 2009 February 13–February 20 | PUVI | 1106.0 |
| | PALO | 1105.0 |
| 2010 March 14–March 21 | PUVI | 951.0 |
| | PALO | 1095.0 |

**Notes.**

[a] PUVI, Pueblo Viejo; PALO, Palo de Aqua.

[b] Net hours, total number of 12-m-equivalent nets × number of hours open.

We did not find that adding the MEI as a covariate improved model fit. The reduction in residual deviance gained by adding MEI as a covariate to the best model was consistently small (0.01–1.0) and always non-significant (all likelihood-ratio test $P$-values > 0.28). The pseudo-$R^2$ of the best model for Green-tailed Ground-Tanager was relatively high (76%), whereas the percent of variation explained by the best model was moderate for Greater Antillean Bullfinch (47%) and low for Rufous-throated Solitaire (30%).

Two other endemics, Hispaniolan Highland-Tanager and Western Chat-Tanager, showed mixed evidence of declines in capture rate. The preferred model for Hispaniolan Highland-Tanager included significant effects of year ($\beta_{year} = -0.23$; 95% CL $= -0.35$, $-0.12$; $P < 0.001$), site ($\beta_{site} = -418.8$; 95% CL $= -687.1$, $168.3$; $P = 0.001$), and their interaction ($\beta_{site*year} = 0.21$; 95% CL $= 0.084$, $0.344$; $P = 0.001$) (Fig. 4). Adding MEI as a covariate did not improve model fit (deviance reduction $= 1.4$, $P = 0.31$). The best model explained 38% of observed variation in capture rate. Declines in expected number

**Table 2  Summary of number of individuals captured.** Number of individuals captured during annual banding sessions conducted at two sites in the Sierra de Bahoruco, Dominican Republic from 1997 to 2010.

| Species | Total individuals captured |
|---|---|
| Sharp-shinned Hawk (*Accipiter striatus*) | 12 |
| White-fronted Quail-Dove (*Geotrygon leucometopia*)[a] | 7 |
| Hispaniolan Parakeet (*Psittacara chloropterus*)[a] | 1 |
| Hispaniolan Emerald (*Chlorostilbon swainsonii*)[a],[b] | 47 |
| Narrow-billed Tody (*Todus angustirostris*)[a] | 140 |
| Hispaniolan Woodpecker (*Melanerpes striatus*)[a] | 22 |
| Hispaniolan Trogon (*Priotelus roseigaster*)[a] | 10 |
| Hispaniolan Pewee (*Contopus hispaniolensis*)[a] | 43 |
| Greater Antillean Elaenia (*Elaenia fallax*) | 29 |
| Rufous-throated Solitaire (*Myadestes genibarbis*) | 126 |
| Bicknell's Thrush (*Catharus bicknelli*) | 149 |
| La Selle Thrush (*Turdus swalesi*)[a] | 22 |
| Red-legged Thrush (*Turdus plumbeus*) | 31 |
| Gray Catbird (*Dumetella carolinensis*) | 1 |
| Ovenbird (*Seiurus aurocapilla*) | 162 |
| Worm-eating Warbler (*Helmitheros vermivorum*) | 4 |
| Black-and-white Warbler (*Mniotilta varia*) | 28 |
| Swainson's Warbler (*Limnothlypis swainsonii*) | 7 |
| Kentucky Warbler (*Geothlypis formosa*) | 1 |
| Common Yellowthroat (*Geothlypis trichas*) | 2 |
| American Redstart (*Setophaga ruticilla*) | 3 |
| Black-throated Blue Warbler (*Setophaga caerulescens*) | 83 |
| Pine Warbler (*Setophaga pinus*) | 1 |
| Hispaniolan Highland-Tanager (*Xenoligea montana*)[a] | 69 |
| Green-tailed Ground-Tanager (*Microligea palustris*)[a] | 245 |
| Banaquit (*Coereba flaveola*) | 4 |
| Black-crowned Palm-Tanager (*Phaenicophilus palmarum*)[a] | 77 |
| Western Chat-Tanager (*Calyptophilus tertius*)[a] | 72 |
| Hispaniolan Spindalis (*Spindalis dominicensis*)[a] | 85 |
| Black-faced Grassquit (*Tiaris bicolor*) | 28 |
| Greater Antillean Bullfinch (*Loxigilla violacea*) | 86 |

**Notes.**
[a] Hispaniolan endemic.
[b] The total number of unique individuals captured is unknown because we could not permanently mark individuals with leg bands.

of captures were predicted for both sites, but the rate of decline was greater at PALO than at PUVI (Fig. 4). At PALO, the expected annual decline was 20.5% (95% CL = 11.7%, 28.8%), while at PUVI it was 2.1% (95% CI = 0.4%–3.9%). However, the significance of these relationships was driven by the exceptionally high capture rate at PALO in 1997. When we excluded data from 1997, none of the regression coefficients, including year ($\beta_{year} = -0.05$; 95% CL = $-0.12$, 0.02; $P = 0.12$), were significantly different from zero.

The situation for Western Chat-Tanager was more complicated, as the preferred model also included significant effects for year ($\beta_{year} = -0.10$; 95% CL = $-0.173$, $-0.026$; $P = 0.008$), site ($\beta_{site} = -245.1$; 95% CL = $-440.2$, $-52.8$; $P = 0.013$), and their interaction ($\beta_{site*year} = 0.122$; 95% CL = 0.026, 0.220; $P = 0.001$), but the interaction was such that expected capture rates declined at PALO while remaining steady or gaining slightly at PUVI (Fig. 5). Expected captures at PALO declined by 9.5% (95% CL = 2.5%, 15.9%)

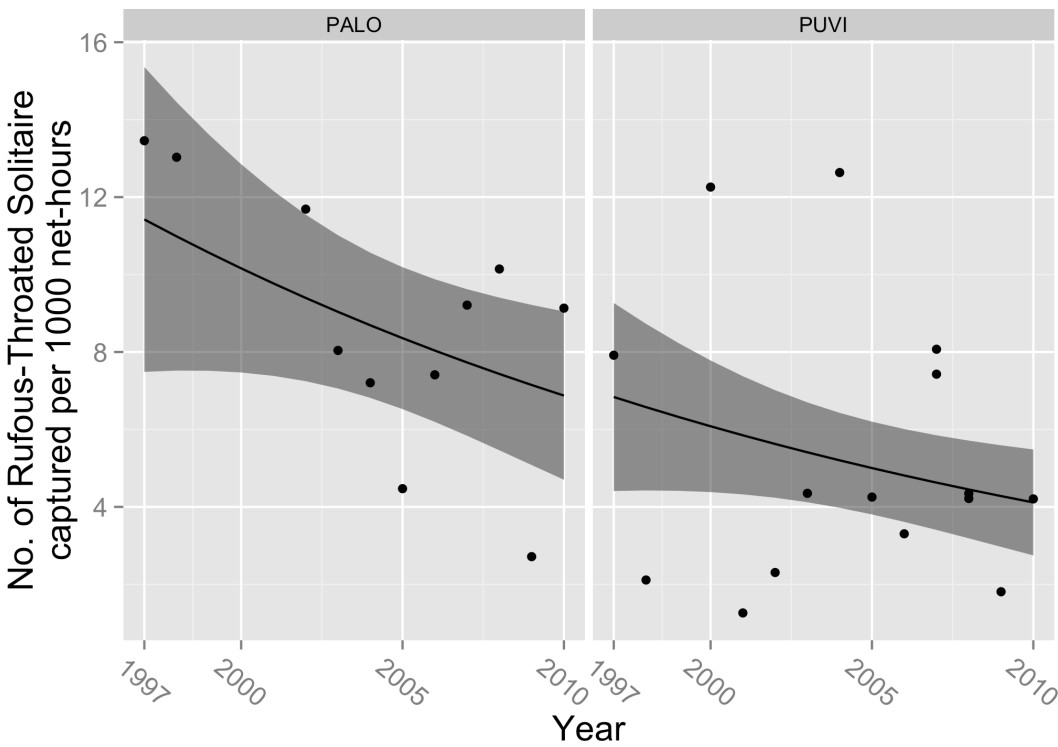

**Figure 1** **Trends in capture rate of Rufous-throated Solitaire (*Myadestes genibarbis*).** Observed capture rate (dots) of Rufous-throated Solitaire and changes in expected capture rate (solid line; shaded area = 95% confidence interval) per 1,000 net-hours at two sites (PALO, Palo de Agua; PUVI, Pueblo Viejo) in montane broadleaf forest of Sierra de Bahoruco, Dominican Republic.

each year, whereas at PUVI expected captures rose by 2.3% per year (95% CL = 0%, 4.8%). However, as with Hispaniolan Highland-Tanager, the statistically significant results were entirely due to the especially high capture rate in 1997; when we excluded that point and re-ran the analysis, none of the regression coefficients differed significantly from zero. Adding MEI as a covariate did not significantly improve model fit, although the effect was stronger than in other species (deviance reduction = 3.5, $P = 0.06$; $\beta_{MEI} = -0.20$; 95% CL = $-0.426, 0.009$; $P = 0.07$). The pseudo-$R^2$ for the best model (site∗year) was 30%.

We found no evidence of any temporal trend in capture rate for Narrow-billed Tody ($\beta_{year} = 0.020$; 95% CL = $-0.017, 0.058$; $P = 0.30$), Black-crowned Palm-Tanager ($\beta_{year} = -0.028$; 95% CL = $-0.079, 0.025$; $P = 0.30$), or Hispaniolan Spindalis ($\beta_{year} = 0.009$; 95% CL = $-0.040, 0.061$; $P = 0.72$), nor did we find evidence for temporal trends in any of the migrant species (Bicknell's Thrush: $\beta_{year} = -0.003$; 95% CL = $-0.042, 0.037$; $P = 0.87$; Ovenbird: $\beta_{year} = 0.004$; 95% CL = $-0.029, 0.039$; $P = 0.78$; Black-throated Blue Warbler: $\beta_{year} = 0.013$; 95% CL = $-0.038, 0.065$; $P = 0.633$). In no case was model fit improved by the inclusion of MEI as a covariate (range of deviance reduction: 0.27–3.1; all likelihood-ratio test $P$-values > 0.09).

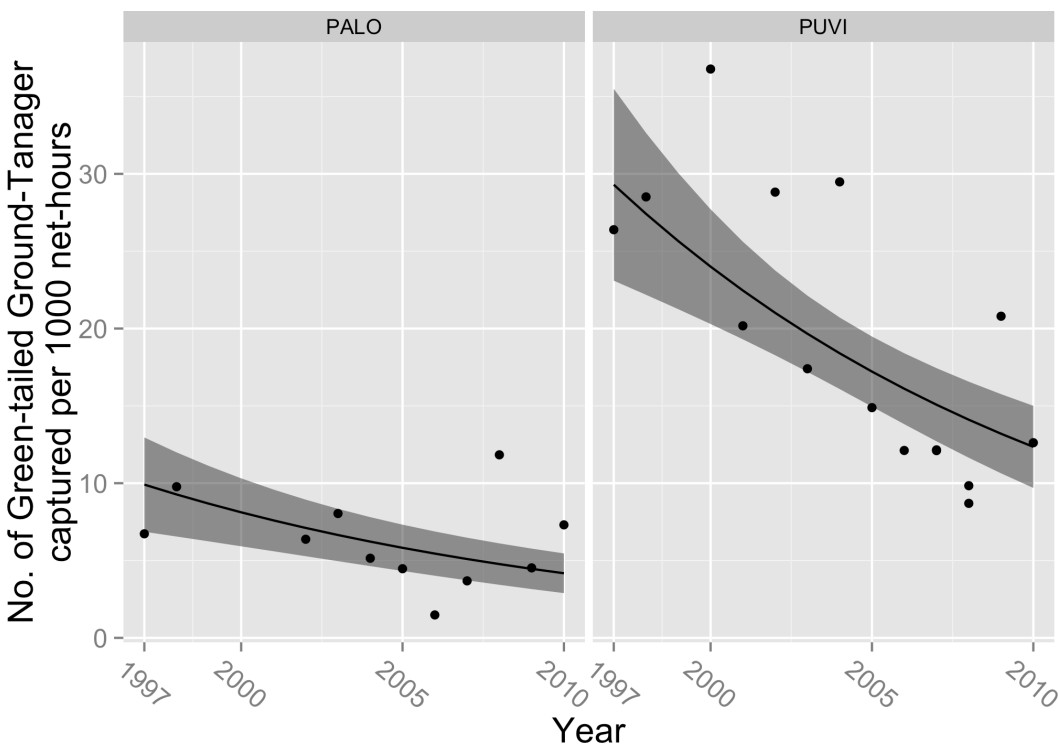

**Figure 2  Trends in capture rate of Green-tailed Ground-Tanager (*Microlegia palustris*).**  Observed capture rate (dots) of Green-tailed Ground-Tanager and changes in expected capture rate (solid line; shaded area = 95% confidence interval) per 1,000 net-hours at two sites (PALO, Palo de Agua; PUVI, Pueblo Viejo) in montane broadleaf forest of Sierra de Bahoruco, Dominican Republic.

## DISCUSSION

Captures rates of three resident species—including one Hispaniolan endemic—declined significantly over the course of this study. Captures of Green-tailed Ground-Tanager declined by ∼63% from 1997 to 2010, Rufous-throated Solitaire by ∼43%, and Greater Antillean Bullfinch by ∼51%. The relationship between capture rate and time was strong for Green-tailed Ground-Tanager, but was relatively weak for the other two species as evidenced by low pseudo-$R^2$ values and substantial scatter in observed capture rates. In addition, the absolute magnitude of the declines was small for Rufous-throated Solitaire and Greater Antillean Bullfinch, amounting to roughly 1–5 fewer birds caught per 1,000 net hours in 2010 as compared to 1997. The magnitude of the decline in Green-tailed Ground-Tanager was greater, with expected capture rates in 2010 ranging from roughly 6 fewer individuals captured per 1,000 net hours at PALO to 17 fewer individuals captured per 1,000 net hours at PUVI. As such, we have relatively more confidence in the biological significance of the modeled decline in capture rate of Green-tailed Ground-Tanager. All of these species are currently considered Least Concern by the IUCN (*BirdLife International, 2012*). However, if the trends that we observed are characteristic of changes occurring range-wide, and if capture rate provides a valid index of population size, then

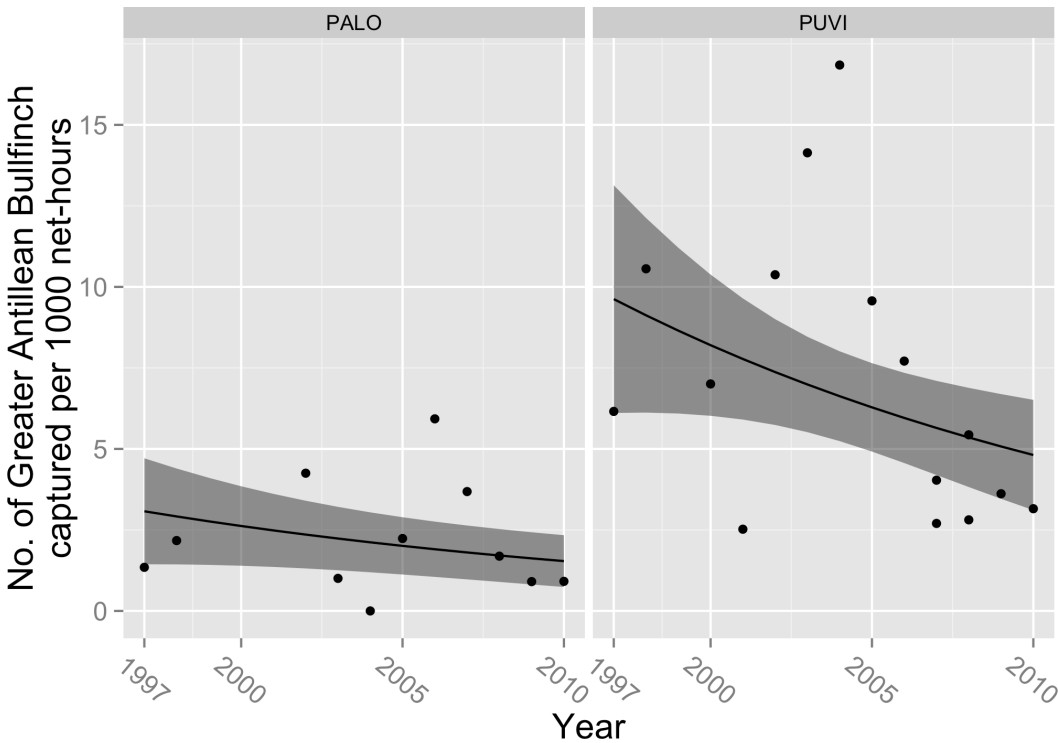

**Figure 3** **Trends in capture rate of Greater Antillean Bullfinch (*Loxigilla violacea*).** Observed capture rate (dots) of Greater Antillean Bullfinch and changes in expected capture rate (solid line; shaded area = 95% confidence interval) per 1,000 net-hours at two sites (PALO, Palo de Agua; PUVI, Pueblo Viejo) in montane broadleaf forest of Sierra de Bahoruco, Dominican Republic.

all of these species would meet the criteria for uplisting to Vulnerable (≥30% decline over 10 years; *IUCN, 2012*).

We have low confidence in estimated trends for two other Hispaniolan endemics, Hispaniolan Highland-Tanager and Western Chat-Tanager. Trends in capture rate varied between sites and were influenced by large numbers of individuals captured in 1997, the first year considered in this analysis. We do not understand why capture rates were so high in 1997, but we are hesitant to conclude that these species declined solely on the basis of results obtained in that year. An equally plausible conclusion is that populations of these two species at our study sites were not in long-term decline, and that data from 1997 reflected an unusual and temporary, if unexplained, increase in the local population available for capture in our nets. Unfortunately, we did not collect information on resident species during our initial visit in 1995, and the only data from 1996 were collected at one site (PUVI) at a different time of year (early December) and so provide little insight into the apparently high capture rates observed in 1997.

Capture rates of the remaining endemics (Narrow-billed Tody, Black-crowned Palm-Tanager, and Hispaniolan Spindalis) were stable during the course of our study. The three migrant species that we examined also showed no trend in capture rate, largely in keeping with concurrent trends estimated on their breeding grounds. Ovenbird and Black-throated Blue Warbler surveys on the breeding grounds indicated a stable to

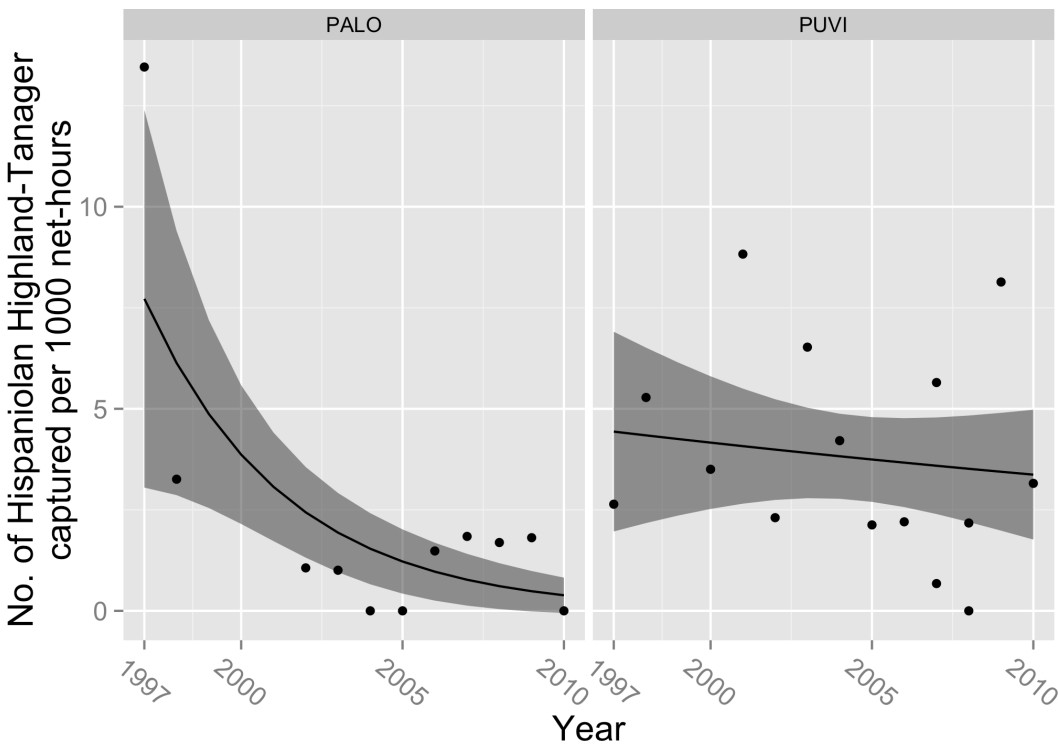

**Figure 4** **Trends in capture rate of Hispaniolan Highland-Tanager (*Xenolegia montana*).** Observed capture rate (dots) of Hispaniolan Highland-Tanager and changes in expected capture rate (solid line; shaded area = 95% confidence interval) per 1,000 net-hours at two sites (PALO, Palo de Agua; PUVI, Pueblo Viejo) in montane broadleaf forest of Sierra de Bahoruco, Dominican Republic.

modestly increasing population over the period of our study (*Sauer et al., 2014*); range-wide estimates of population trend are not available for Bicknell's Thrush, although local declines have been noted (*Lambert et al., 2008*).

We can only speculate about the causes of observed declines. We saw no clear suggestion that declines were related to climate or weather. Reduced food availability mediated by reduced precipitation during warm-phase ENSO events can limit survival of migrant and resident birds in the Neotropics (*Sillett, Holmes & Sherry, 2000*; *Wolfe, Ralph & Elizondo, 2015*), but we found no evidence of a relationship between MEI and capture rate for any species. The intact montane forest that characterized our study sites may be resistant to ENSO-driven variability in precipitation (e.g., *Wolfe, Ralph & Elizondo, 2015*), but the lack of any clear signal of ENSO may also reflect variability in the effect of ENSO on local and regional precipitation patterns (e.g., *Jury, Malmgren & Winter, 2007*). Hurricanes can have profound effects on bird populations in the Caribbean (*Waide, 1991*; *Wiley & Wunderle, 1993*), but we saw no obvious relationship between the passage of tropical cyclones through our study sites and changes in capture rate. For example, Hurricane Georges, the most powerful cyclone to affect our study area during the course of this research, caused widespread damage across the Dominican Republic and passed almost directly over Sierra de Bahoruco in September 1998. We did not visit either study site in 1999, but capture rates for most species were relatively high in 2000,

 

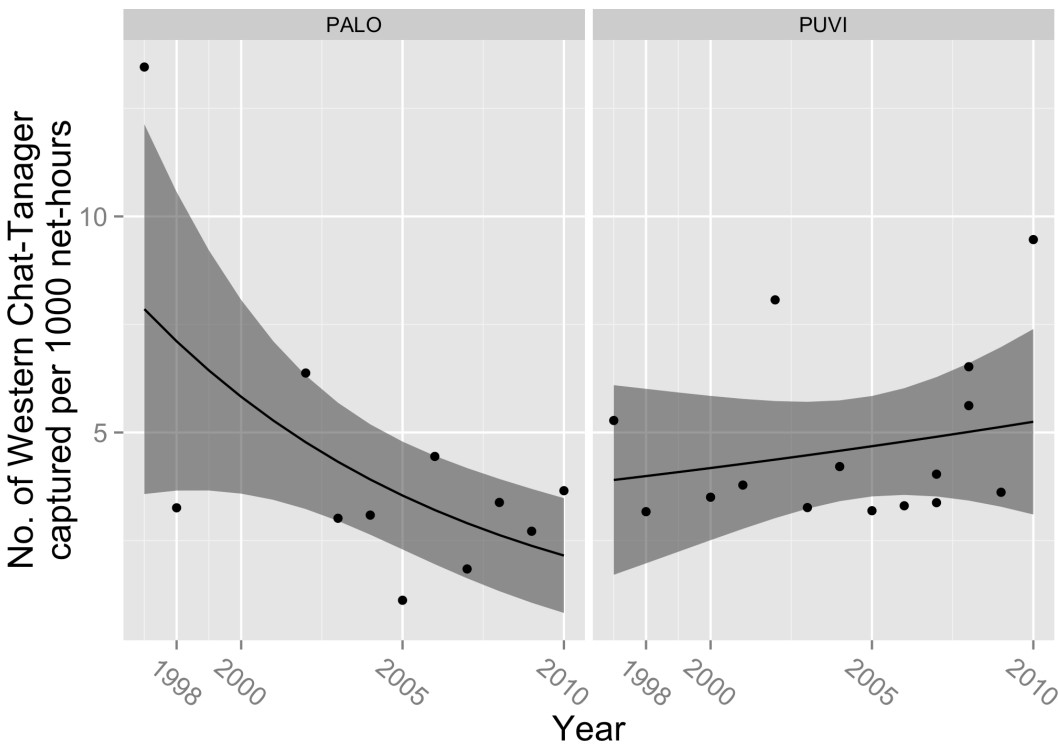

**Figure 5** **Trends in capture rate of Western Chat-Tanager (*Calyptophilus tertius*).** Observed capture rate (dots) of Western Chat-Tanager and changes in expected capture rate (solid line; shaded area = 95% confidence interval) per 1,000 net-hours at two sites (PALO, Palo de Agua; PUVI, Pueblo Viejo) in montane broadleaf forest of Sierra de Bahoruco, Dominican Republic.

even among those species that showed long-term declines in capture rate. If hurricane-related changes in habitat conditions were responsible for the declines that we observed, then we would have expected a sharp drop in capture rate after 1998. We also suspect little role for local changes in vegetation structure or composition. Anthropogenic effects on vegetation structure at the sites were minimal and restricted to our maintenance of net lanes. Surrounding forests were also largely free from direct, human-caused disturbance. Natural disturbances were limited to a few small, tree-fall canopy gaps, and we consider it unlikely that patchy successional changes contributed substantially to any long-term declines in capture rate.

Both study sites support large populations of introduced rats (*Rattus rattus* and *R. norvegicus*), which are probably important predators of adult birds and nests (*Townsend et al., 2009*), but why they would affect only certain species is unclear. Also unclear is why rats, which have been established on most islands of the Caribbean for several hundred years (*Harper & Bunbury, 2015*), including presumably Hispaniola, would precipitate recent declines. Finally, habitat loss outside of the study area caused by extensive, ongoing deforestation (*BirdLife International, 2015*) could drive local declines by reducing the regional population and thus reducing both recruitment into local populations and the number of transient individuals exposed to our sampling efforts.

Capture rate in mist nets is often a valid index of abundance, and trends in capture rate are usually—but not always—correlated with population trends estimated using other methods (*Dunn & Ralph, 2002*). However, changes in capture rate might also reflect changes in our ability to capture individuals, rather than changes in the number of individuals available for capture. Although capture probability need not be constant, the validity of our inferences regarding changes in abundance over time requires that variation in capture probability was less than variation in abundance (*Johnson, 2008*). We controlled for variation in effort and held constant other factors that might influence capture rate independently of abundance, such as net location, mesh size, vegetation structure immediately around the nets, and seasonal timing of capture efforts. As with any study that uses capture rates as an index of abundance, we cannot rule out the possibility that individual birds learned to avoid our net locations, which could lead to changes in capture rate independent of changes in abundance. However, we believe that net avoidance is an unlikely explanation for the observed declines, given that we opened nets only for several days per year.

Assuming that trends in capture rate reflected trends in the number of individuals available for capture, how might these findings inform assessments of conservation status? Whether the trends described here were purely a local phenomenon is uncertain. We sampled a small number ($n = 2$) of purposefully selected sites (undisturbed by human activity) within the montane cloud forest of Sierra de Bahoruco, and so we cannot use these data to draw inference more broadly about the range-wide status of any species. Nonetheless, when combined with other sources of information on population trends, our findings are useful in highlighting which species warrant closer scrutiny. For example, Green-tailed Ground-Tanager, Rufous-throated Solitaire, and Greater Antillean Bullfinch have been reported as declining in other recent evaluations of conservation status (*Latta et al., 2006*; *BirdLife International, 2012*), which suggests, but does not demonstrate, that the patterns we described may not be limited solely to our study areas.

Even with these findings, which represent the only long-term, quantitative information available on population trends for these species, substantial uncertainty remains regarding range-wide patterns of vulnerability. Intensive studies like ours can provide useful information about ecology and local demographics, but are limited in the scope of inference that they allow regarding overall changes in population parameters. Given the challenges of funding and executing geographically extensive biodiversity monitoring studies, especially in the tropics, it is unlikely that any additional quantitative information can be collected in the short-term that would help resolve this uncertainty. As such, a useful next step might be to focus on other criteria important in assessing vulnerability (*IUCN, 2012*), such as extent of occurrence, that could be quantified using currently available data (e.g., species distribution modeling using data from sources like eBird) but which are not well described at present.

## ACKNOWLEDGEMENTS

Field assistance was provided by many local and international partners, but several deserve special mention: J Almonte, E Garrido, J Goetz, J Klavins, R Ortiz, and J Townsend.

### Funding

Funding for our work over the years was provided by multiple sources, including the Carolyn Foundation, MacArthur Foundation, The Nature Conservancy, Stewart Foundation, Thomas Marshall Foundation, US Fish and Wildlife Service, US Forest Service International Program, and friends of both the Vermont Center for Ecostudies and Vermont Institute of Natural Science. The funders had no role in study design, data collection and analysis, decision to publish, or preparation of the manuscript.

### Grant Disclosures

The following grant information was disclosed by the authors:
Carolyn Foundation.
MacArthur Foundation.
The Nature Conservancy.
Stewart Foundation.
Thomas Marshall Foundation.
US Fish and Wildlife Service.
US Forest Service International Program.
Vermont Center for Ecostudies.
Vermont Institute of Natural Science.

### Competing Interests

The authors declare there are no competing interests.

### Author Contributions

- John D. Lloyd conceived and designed the experiments, performed the experiments, analyzed the data, contributed reagents/materials/analysis tools, wrote the paper, prepared figures and/or tables, reviewed drafts of the paper.
- Christopher C. Rimmer and Kent P. McFarland conceived and designed the experiments, performed the experiments, wrote the paper, reviewed drafts of the paper.

### Animal Ethics

The following information was supplied relating to ethical approvals (i.e., approving body and any reference numbers):

Permission to band North American migrants was granted by the USGS Bird Banding Lab, under a permit issued to CCR (permit no. 23,541), and research activities in the Dominican Republic were approved by the Subsecretaria de Áreas Protegidas y Biodiversidad.

## Data Availability

KNB Data Repository. http://dx.doi.org/10.5063/F1M906K7.

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
