# Peer review of "Assessing conservation status of resident and migrant birds on Hispaniola with mist-netting"

_PeerJ, doi:10.7717/peerj.1541_

## Round 0.1 · original submission · Major Revisions

In addition to the 2 reviews, please also find the manuscript with my own comments attached. While both reviewers suggested minor revisions, for the reasons I write in my comments in your manuscript, I think the revision should be a bit more extensive. My first and foremost concern was that your study (which is great that you have data over the long haul) is a typical capture-mark-recapture study. As such, using mark and recapture statistics, all available in the program MARK, you can specifically test time and estimate both survival and capturability over time. And, because your data are not temporally independent a GLM approach might be inappropriate. Some figures suggest that temporally your data are not independent.

One alternative that you did not address in your study is whether birds learn to avoid mistnets. If so, then each year, you would expect that recaptures decline and you tend to capture new birds each time. Learned avoidance of mistnets could be the entire explanation for your trend over time. If birds become net shy, then your probability of recapture will decrease with the number of times each bird is captured. You did not specifically address this alternative possibility. In your text (around lines 270 on page 14, for example), you speculate on the causes of the declines but never consider net avoidance or declining capturability. The program MARK can allow you to specifically test those ideas.

A (perhaps) minor detail is also that you state that captures decline when those declines are not statistically significant. If a trend is not significant, then it is NOT a trend and cannot be stated as such. I felt that you might be overstating your case, and I am reasonably sure that with Jolly-Seber type analyses, you will be able to state that case more clearly.

One reviewer did not provide much in the way of constructive criticism, and the other only pointed out some minor details, of which some are also noted in my comments within your paper.

Reviewer 1 ·

Basic reporting

No comment

Experimental design

Basic design was fine. Would be interesting if survival estimates were possible, but low capture numbers for most species most likely prevents calculating accurate estimates.

Validity of the findings

The authors did not try to overstate any results. They very deliberately stated the dearth of information available renders any trends informative.

Additional comments

After carefully reading this manuscript, I found only one problem, an incomplete sentence ending on line 306. The paper was clearly written. The lack of information for Caribbean species, and particularly island endemics makes results from all carefully conducted studies important.

·

Basic reporting

There are some incorrect Latin names and others don't follow the current taxonomy. Besides, the common names should follow a source accepted worldwide (e.g. IUCN, ). I recommend the alterations bellow:
Microlegia palustris (wrong spelling) should be Microligea palustris
Xenolegia montana (wrong spelling) should be Xenoligea montana
Aratinga chloroptera(old nomenclature) should be Psittacara chloropterus
Priotelus roseigaster (old nomenclature) should be Temnotrogon roseigaster
Geothlypis formosa (old nomenclature) should be Oporornis formosus
Setophaga caerulescens (old nomenclature) should be Dendroica caerulescens
Setophaga pinus (old nomenclature) should be Dendroica pinus

Line 273: the citation Waide 1981 is referenced as Waide RB. 1991 in the line 411

Experimental design

Line 61: it is defined only one type of net, but in the lines 146-148 other nets are mentioned, this should be clarified.

Lines 96-98: the sentence "... although at both sites we also occasionally recorded captures at locations where nets were deployed opportunistically for related research projects..." should be better explained. I didn't understand what exactly was done.

Validity of the findings

In the lines 98-100 it is said that the effort per site per year was ~3 days. But in Table 1 it is possible to see great differences in the amount of net-hours per site in each year. Although the captures were considered as an index, we know that it is expected a depletion on capture during each session due trap-shy behavior of some species. I think that this decrease in the captures isn't linear; with more effort the decrease of the captures will be great. So, the indexes of captures based on different efforts should be biased. I would like to see a model considering as a predictor variable the net-hours of each year of the Table 1. Maybe, the effort per year-site in net-hours can be a better predictor. At least, the authors should discuss this issue in the discussion and justify why this is not a problem.

Additional comments

Congratulations for the long term monitoring program developed. I know that it is very difficult to maintain a research for a long time. We need long series of data to be more conclusives, unfortunately nowadays the researches are, in general, of only a couple of years.

---

## Round 0.2 · Minor Revisions

This new manuscript is improved, but I still have a variety of comments, most of which I would think other readers would also have. Please see my attached document containing those comments. That document is a reply to your rebuttal.

---

## Round 0.3 · accepted · Accept

I appreciate your extensive reply and I agree, in spirit let's say, with your perspective on generating discussion and using the data in the best way possible. Regardless of minor points of debate about the details, it is important that we get information pertinent to conservation and ecology for relatively poorly know, but potentially endangered species.